# Comparison between Cervical *Ureaplasma* spp. Colonization and the Intensity of Inflammatory Mediators in the Amniotic Fluid Retrieved during Cesarean Delivery in Preterm Birth

**DOI:** 10.3390/ijerph19010107

**Published:** 2021-12-23

**Authors:** Jingon Bae, Shin Kim, Ilseon Hwang, Jaehyun Park

**Affiliations:** 1Department of Obstetrics and Gynecology, Keimyung University School of Medicine, Daegu 42601, Korea; jgonmd@gmail.com; 2Department of Immunology, Keimyung University School of Medicine, Daegu 42601, Korea; god98005@dsmc.or.kr; 3Institute for Medical Science, Keimyung University, Daegu 42601, Korea; ilseon@dsmc.or.kr; 4Department of Pathology, Keimyung University School of Medicine, Daegu 42601, Korea; 5Department of Pediatrics, Keimyung University School of Medicine, Daegu 42601, Korea

**Keywords:** *Ureaplasma*, chorioamnionitis, amniotic fluid, preterm birth

## Abstract

We investigated whether cervical *Ureaplasma* spp. colonization affects the intensity of inflammatory mediators in amniotic fluid retrieved during cesarean delivery in singleton preterm birth. One hundred fifty-three cases in singleton preterm birth with 24–34 weeks’ gestation were enrolled. The intensities of seven inflammatory mediators (interleukin (IL)-1β, IL-6, IL-8, IL-10, tumor necrosis factor-α, and matrix metalloproteins (MMP)-8, MMP-9) of amniotic fluid were measured. We tested cervical swab specimens using real-time polymerase chain reaction assays to detect *Ureaplasma* spp. colonization. Histologic chorioamnionitis (HCA) was diagnosed when acute inflammation was observed in any of the placental tissues. Mean gestational age at delivery and birth weight were 30.9 ± 2.4 weeks and 1567 ± 524 g, respectively. Cervical *Ureaplasma* spp. colonization was detected 78 cases. The incidence of HCA was 32.3% (43/133). Although the intensities of all inflammatory mediators were significantly different according to presence or absence of HCA, there were no significant differences according to cervical *Ureaplasma* spp. colonization. In all 43 cases with HCA and 90 cases without HCA, there were no significant differences between cervical *Ureaplasma* spp. colonization and the intensity of inflammatory mediators. Cervical *Ureaplasma* spp. colonization did not affect the intensity of inflammatory mediators in the amniotic fluid retrieved during cesarean delivery.

## 1. Introduction

Preterm birth occurs in approximately 10% of all pregnancies worldwide and is a major contributor to perinatal mortality and adverse outcomes [1]. Chorioamnionitis or intraamniotic infection accounts for 25–40% of preterm birth [2,3,4]. According to recommendations of the American College of Obstetricians and Gynecologists, chorioamnionitis, which is defined as an infection with resulting inflammation of any combination of amniotic fluid, placenta, fetal membranes, or decidua, can be diagnosed by amniotic fluid culture, gram stain, or both accompanied with biochemical analysis [5]. Chorioamnionitis occurs mainly as an ascending bacterial infection in rupture of the amniotic membrane but can also occur in intact membranes [6,7,8,9]. Nevertheless, chorioamnionitis is often asymptomatic and occurs without microbiologically proven amniotic fluid infections [3,7,10]. The microbial invasion into the chorioamnion engenders a maternal and fetal inflammatory response characterized by the release of proinflammatory or inhibitory cytokines and chemokines [6]. Accumulating evidence suggests that an intraamniotic inflammatory response is associated with a variety of dysregulated mediators such as interleukin (IL)-1 ß, IL-6, IL-8, IL-10, tumor necrosis factor (TNF)-α), matrix metalloproteinase (MMP)-8, and MMP-9 [7,8,9,11,12,13,14].

Among the pathogenic bacteria associated with intraamniotic infections, *Ureaplasma* spp. is the most common organism that is isolated from an infected amniotic fluid and a placenta and is associated with an increased risk for preterm labor and perinatal morbidity [15,16,17,18,19]. *Ureaplasma* spp., which colonizes the lower genitalia, penetrates into the choriodecidual space, causing preterm labor. According to recent reports, *Ureaplasma* spp. is a commensal organism of the female lower genital tract that demonstrates no difference in the rates of colonization between women with and without symptomatic genital infection [20,21]. In particular, *Ureaplasma* spp.-associated inflammation within the chorioamnion may vary between 0 and 100% [22]. A causative role between *Ureaplasma* spp. colonization in the lower genital tract and adverse outcomes during pregnancy remains controversial. Therefore, in the present study, we investigated whether cervical *Ureaplasma* spp. colonization affects the intensity of arbitrarily selected inflammatory mediators in the amniotic fluid retrieved during cesarean delivery in singleton preterm births.

## 2. Materials and Methods

### 2.1. Collection of the Amniotic Fluids

Before the collection of the amniotic fluid, an agreement with the Keimyung Human Bio-Resource Bank was made, and an approval from the National Biobank of Korea was received [23,24]. Informed consent was acquired to collect the amniotic fluid during cesarean delivery from pregnant women with threatened preterm birth between 24 and 34 weeks of gestation. After the administration of spinal anesthesia to the pregnant women, a low transverse cesarean incision was made, and hysterotomy was performed. The myometrium was incised with shallow strokes to avoid amniotic sac rupture. After exposing amniotic sac (Figure 1), amniocentesis was performed using a 21-gauge needle. To reduce any further risk to the pregnant women and their babies during delivery, amniocentesis was not performed in patients who were placed under general anesthesia, who had severe oligohydramnios (amniotic fluid index <1.0), with suspected placental abruption with fetal distress, or with unexpected amniotic sac rupture during cesarean delivery. The collected amniotic fluid was centrifuged, and the supernatant was aliquoted and stored at −80 °C at the Keimyung Human Bio-Resource Bank. Samples were not subjected to freeze–thaw cycles before being assayed. Amniotic fluids between January 2017 and December 2019 were provided by the Keimyung Human Bio-Resource Bank. This study was approved by the Dongsan Medical Center Institutional Review Board (approval No. DSMC 2020-01-001).

### 2.2. Inclusion and Exclusion Criteria

This observational cohort study was conducted at the High-Risk Maternal and Newborn Integrated Care Center, Keimyung University Dongsan Hospital, Daegu, Republic of Korea. Clinical information of pregnant women whose amniotic fluid was collected during cesarean delivery from January 2017 to December 2019 was investigated using medical records, including information on cervical *Ureaplasma* spp. colonization and histologic chorioamnionitis (HCA).

We also investigated other clinical characteristics, including maternal age; parity; any history of cerclage intervention, gestational diabetes, cervical dilatation at admission, preterm labor with intact membrane, preterm premature rupture of membrane (PPROM), pregnancy-induced hypertension (PIH), placenta previa, placenta abruptio, and oligohydramnios; gestational age at delivery; and birth weight. Gestational age was estimated on the basis of the mother’s last menstrual period and ultrasonography findings. Cerclage intervention was defined as a prophylactic operative intervention to treat painless cervical dilation in the second trimester. Preterm labor was defined as regular contractions of the uterus resulting in cervical dilatation. PPROM was defined as the rupture of membranes earlier than 24 h before the onset of labor.

### 2.3. Amniotic Fluid Analysis

Seven arbitrarily selected inflammatory mediators (IL-1β, IL-6, IL-8, IL-10, TNF-α, MMP-8, and MMP-9) of the amniotic fluid were quantitatively measured using a Human Magnetic Luminex screening assay (R&D Systems, Minneapolis, MN, USA) on the Bio-Plex 200 Sysytems (Bio-rad Laboratories, Hercules, CA, USA). All the measurements were carried out strictly according to the manufacturer’s instructions, and all samples were measured in duplicate at the same time. For the IL-1β assay, a standard curve was developed from 39.0 to 9488.0 pg/mL with a sensitivity of 1.6 pg/mL; for the IL-6 assay, the curve was linear from 9.6 to 2308.0 pg/mL with a sensitivity of 3.4 pg/mL; for the IL-8 assay, the curve was linear from 10.4 to 2510.0 pg/mL with a sensitivity of 3.6 pg/mL; for the IL-10 assay, the curve was linear from 9.6 to 2324.0 pg/mL with a sensitivity of 3.2 pg/mL; for the TNF-α assay, the curve was linear from 19.4 to 4718.0 pg/mL with a sensitivity of 2.4 pg/mL; for the MMP-8 assay, the curve was linear from 490.2 to 119,124.0 pg/mL with a sensitivity of 68.4 pg/mL; and for the MMP-9 assay, the curve was linear from 6705.0 to 1,629,800.0 pg/mL with a sensitivity of 680.0 pg/mL.

### 2.4. Pathological Investigation of Placenta

Placentas were subjected to histologic evaluation; representative sections included the chorioamnion, chorionic plate, and umbilical cord. These samples were fixed in 10% neutral-buffered formalin and embedded in paraffin. Sections of tissue blocks were stained with hematoxylin and eosin. HCA was diagnosed when acute inflammation was observed in any of the placental tissues.

### 2.5. Detection of Cervical Ureaplasma *spp.* Colonization

Investigations for microorganisms in the lower genital tract are routinely performed in our institution when a pregnant woman with a threatened preterm delivery is hospitalized. The detection of cervical *Ureaplasma* spp. colonization in the cervix was performed using Anyplex II Sexually Transmitted Infections-7 Kit (STI-7 Seegene, Seoul, Republic of Korea), a commercially available multiplex real-time polymerase chain reaction (PCR) relying on newly developed Tagging Oligonucleotide Cleavage and Extension technology, which allows for the simultaneous detection of seven microorganisms (*Ureaplasma urealyticum, Ureaplasma parvum*, *Mycoplasma hominis*, *Mycoplasma genitalium*, *Chlamydia trachomatis*, *Neisseria gonorrhoeae*, and *Trichomonas vaginalis*).

### 2.6. Statistical Analysis

Before the statistical analysis, values of the inflammatory mediators were log2 transformed. Continuous variables were expressed as means ± standard deviation and categorical variables as numbers and proportions. The clinical characteristics and the intensity of the seven inflammatory mediators were compared according to the presence or absence of HCA. Comparisons between categorical variables were performed using the chi-square test or Fisher’s exact test, and those between continuous variables were performed using the independent t-test. Previous studies reported that the earlier the gestational age, the more intense the intraamniotic inflammation in women with preterm labor and PPROM [8,9,12,25,26]. Considering the aforementioned studies, we investigated the Pearson correlation coefficient between gestational age at delivery and the inflammatory mediators of the amniotic fluid. Finally, a multivariate logistic regression analysis was conducted to investigate whether cervical *Ureaplasma* spp. colonization was an independent risk factor for HCA, adjusted by clinical factors that showed significant differences according to the presence or absence of HCA. The statistical analysis was performed using SPSS version 25.0 (IBM Co., Armonk, NY, USA). A *p* value of <0.05 was considered statistically significant.

## 3. Results

As shown in Figure 2, microorganisms were detected in 93 cases (62.7%) of the 153 cases of singleton preterm births. Twenty cases were colonized with two or more microorganisms including *Ureaplasma* spp. A single strain of *Ureaplasma* spp. was detected in 78 cases; *U. parvum* and *U. urealyticum* were 59 cases and 19 cases, respectively.

Of the 133 total cases included for this analysis, HCA was present in 43 cases (32.3%), while cervical *Ureaplasma* spp. colonization was present in 78 cases (58.6%). The mean gestational age at delivery and birth weight were 30.9 ± 2.4 weeks and 1567 ± 524 g, respectively.

Table 1 shows the comparison of clinical characteristics and mean values of inflammatory mediators according to the presence or absence of HCA. At first, the incidence of cervical *Ureaplasma* spp. colonization in the HCA (+) group was not significantly different than that in the HCA (−) group (60.5% (26/43) vs. 57.8% (52/90), *p* = 0.768). In addition, the mean gestational age at delivery in the HCA (+) group was significantly lower than that in the HCA (−) group (30.2 ± 2.5 vs. 31.4 ± 2.2 weeks, *p* = 0.008). Preterm labor with intact membrane and PPROM rates in the HCA (+) group were significantly higher than those in the HCA (−) group (51.2% (22/43) vs. 26.7% (24/90), 37.2% (16/43) vs. 16.7% (15/90), respectively, *p* < 0.05). Cervical dilatation at admission in the HCA (+) group was significantly more advanced than that in the HCA (−) group (1.5 ± 1.8 vs. 0.8 ± 1.3 cm, *p* = 0.018). Meanwhile, the incidence of PIH in the HCA (+) group was significantly lower than that in the HCA (−) group (25.6% (11/43) vs. 50.0% (45/90), *p* = 0.008). However, there were no differences in the preterm labor with intact membrane and PPROM rates between the two groups. In the comparison of inflammatory mediators, all mediators in the HCA (+) group were significantly increased (*p* < 0.001).

Figure 3 shows the comparison between the mean values of intraamniotic inflammatory mediators and cervical *Ureaplasma* spp. colonization. There was no difference in all intraamniotic inflammatory mediators according to the presence or absence of cervical *Ureaplasma* spp. colonization. In all 43 cases with HCA and 90 cases without HCA, there were no significant differences between the cervical *Ureaplasma* spp. colonization and the intensity of intraamniotic inflammatory mediators.

Table 2 shows the correlation between intraamniotic inflammatory mediators and gestational age at delivery. Except for IL-8, other inflammatory mediators demonstrated weakly negative linear relationships with gestational age at delivery.

Figure 4 shows the comparison of intraamniotic inflammatory mediators between cervical *U. parvum* and *U. urealyticum*. colonization. *U. parvum* and *U. urealyticum* were present in 59 and 19 cases, respectively. There were no significant differences in the intensity of intraamniotic inflammatory mediators.

Table 3 shows the independent risk factors associated with HCA based on the multivariate logistic regression analysis. Cervical *Ureaplasma* spp. colonization was not an independent risk factor of HCA. The adjusted odds ratios (95% confidence interval: 95% CI) of preterm labor with intact membrane, PPROM, and gestational age at delivery were 5.041 (1.985–12.800), 7.217 (2.559–20.357), and 0.776 (0.651–0.926), respectively.

## 4. Discussion

In the present study, we investigated the effect of cervical *Ureaplasma* spp. colonization on the intensities of inflammatory mediators in the amniotic fluid retrieved at cesarean delivery. We showed that cervical *Ureaplasma* spp. colonization was not significantly associated with the incidence of HCA in singleton preterm births with 24–34 weeks’ gestation. Based on the quantitative analysis of inflammatory mediators, cervical *Ureaplasma* spp. colonization did not affect the intensity of intraamniotic inflammatory mediators in patients with or without HCA.

Spontaneous preterm birth is thought to be due to chorioamnionitis in 25–40% of cases [2,3,4]. The most common pathogens are *Ureaplasma* spp. and *M. hominis* [19,27,28,29]. *Ureaplasma* spp. colonization may result in preterm labor through the production of cytokines that initiate contractions [30]. In addition, cervical *Ureaplasma* spp. colonization has been associated with increased levels of inflammatory mediators, which are associated with precipitation of preterm labor and PPROM [31]. We believed that it was necessary to compare the relationship between the intensity of intraamniotic inflammatory mediators and cervical microbial investigation using a multiplex real-time PCR, which is relatively common to use in clinical practice. In the present study, *Ureaplasma* spp. was detected in 62.7% of patients undergoing multiplex real-time PCR. On the contrary, *Mycoplasma* spp. were detected in 7.8%. Here, 20 cases with two or more microbial colonization in the cervix were excluded to avoid the skewed outcomes by other microorganisms colonized in the cervix. Meanwhile, a culture-based investigation of the amniotic fluid was only performed when chorioamnionitis was suspected; therefore, culture results of the amniotic fluid could not be pooled from all enrolled patients.

According to previous studies, positive [32,33] and negative [34,35,36] results coexist in regard to the role of cervical *Ureaplasma* spp. colonization as a risk factor for chorioamnionitis and preterm birth [33,34]. In this study, cervical *Ureaplasma* spp. colonization was not an independent risk factor of HCA in preterm birth and did not affect the intensity of intraamniotic inflammation in a comparison to those with and without HCA. Similarly, previous reports considered *Ureaplasma* spp. to be a commensal organism [20,21]. However, we only investigated *Ureaplasma* spp. colonization through a cervical swab analysis without considering intraamniotic *Ureaplasma* spp. infections. A study comparing *Ureaplasma* spp. colonization in the amniotic fluid or placental tissue would be helpful to elucidate further results.

Multifactorial etiologies may result in iatrogenic preterm birth in women who have multiple pregnancies [37]. Various pathophysiologies include intrauterine inflammation, cervical insufficiency, uterine overdistension, hormonal disorders, uterine ischemia, and abnormal allograft reaction [38]. In this study, only singleton preterm births were included, and patients with multiple pregnancy were excluded. A study comparing the difference in the intensity of intraamniotic inflammatory mediators and microbiological strains among each fetus in multiple pregnancies may be necessary.

Although the pathophysiology of PIH is still not entirely understood, the development of PIH involves an inflammatory imbalance during placental invasion that leads to endothelial abnormalities [39,40]. Socha et al. reported that the NLRP3 inflammasome, which is involved in production of IL-1 ß and IL-18, plays a critical role in the development of PIH [40]. However, in this study, the incidence of PIH in the HCA (+) group was significantly lower than that in the HCA (−) group. Although further research is needed, it can be said that there is a limit to explaining PIH by only the increase of NLRP3 inflammasome-related IL-1 ß.

Labor, which is characterized by cervical ripening and uterine contraction, is considered as a sterile inflammatory event [41]. The major changes in proinflammatory and anti-inflammatory cytokine expression in the cervix occur during the labor process irrespective of gestational age [42,43]. In particular, Reyes-Lagos et al. reported that the systemic downregulation of several members of the IL-10 family plays an important role in the uterine contraction during active labor [43]. Further studies on the immunological mechanisms including the IL-10 family for term or preterm labor are needed. Meanwhile, previous studies reported that labor was associated with increased risk of microbial invasion of the amniotic cavity, intraamniotic inflammation, and HCA [44,45]. This could be explained by the mechanism that uterine contraction has a suction-like effect and that microorganisms were transferred by digital examination to check the progress of labor [44]. Moreover, the intensity of host response to microbial invasion of the amniotic cavity is greater in the context of preterm than term labor [45]. In this study, cervical dilation at admission in the HCA (+) group was also more advanced in singleton preterm births with 24–34 weeks’ gestation.

There was an inversely proportional relationship between the intensity of intraamniotic inflammatory mediators and gestational age [8,9,12,25,26]. In this study, the mean gestational age at delivery in the HCA (+) group was lower than that in the HCA (−) group. Furthermore, except for IL-8, other inflammatory mediators demonstrated a weakly negative linear relationship with gestational age at delivery. These results are considered to be a major factor that must be corrected to compare the intensity of intraamniotic inflammation.

Similar to the findings in previous studies [20,21], *U. parvum* was more frequent in the lower genital tract than *U. urealyticum*. The clinical difference between *U. parvum* and *U. urealyticum* in the lower urogenital tract of females remains unclear. In this study, there were no significant differences in the intensity of intraamniotic inflammatory mediators between *U. parvum* and *U. urealyticum*.

A particular strength of the present study is that the amniotic fluid was extracted during cesarean delivery. In previous studies, the amniotic fluid was collected via an ultrasound-guided transabdominal approach before delivery [7,8,9,11,12,13,46]. Although ultrasound guidance for amniocentesis is not associated with amniocentesis-attributable complications, such as preterm labor, placental abruption, PPROM, and fetal heart rate abnormality [47], the possibility of emergent cesarean delivery after transabdominal amniocentesis is a major complication to be considered during the third trimester [48,49]. The method of collecting the amniotic fluid during cesarean delivery is not entirely new. According to a previous study by Dudley et al. [50], a portion of the amniotic fluid was collected at the time of hysterotomy during cesarean delivery; however, it was difficult to interpret differences in the intensity of inflammatory mediators in the amniotic fluid collected through various pathways. In another report by Seong et al. [44], the amniotic fluid was retrieved by aspiration under direct visualization after uterine incision at the time of cesarean section in pregnant full-term women with an intact membrane at more than 37 weeks’ gestation. Therefore, a notable distinction of our study was that all the amniotic fluid was collected for preterm births at 24–34 weeks’ gestation collected similarly during cesarean delivery. Since the amniotic fluid was extracted by puncture under direct visualization only when the amniotic sac was exposed, the amniotic fluid was also collected in PPROM-exposed cases.

The present study has several limitations. First, the gold standard for the diagnosis of an intraamniotic infection is a positive microbiologic culture for microorganisms; thus, a PCR test or *Ureaplasma* spp. cultivation techniques should be performed on the amniotic fluid. Yoon et al. [51] reported that a PCR assay for *U. urealyticum* in amniotic fluid samples showed in a higher detection rate than standard microbiologic cultures. *Ureaplasma* spp. is a fastidious microorganism that is difficult to culture, requiring a special metabolic substrate and growth factor for isolation. The detection of *Ureaplasma* spp. required commercial diagnostic kits using PCR [30]. Second, although this study was conducted by a single tertiary institution, it was difficult to investigate the indication, type, and duration of antibiotics administration in pregnant women who were recruited from secondary medical institutions. It may be necessary to compare whether antibiotic treatment for *Ureaplasma* spp. infection was administered and if chorioamnionitis was suspected or not. Third, other microorganisms including *Gardnerella vaginalis*, *Group B Streptococcus*, and *Candida albicans* were not investigated. Fourth, we analyzed amniotic fluid using a Human Magnetic Luminex screening assay for investigating whether cervical *Ureaplasma* spp. colonization affects the intensity of the inflammatory mediators, not by immunohistochemistry or reverse transcription–polymerase chain reaction. These results have limitations in analyzing the causative tissues that secret inflammatory mediators. Additional studies with larger population sizes are needed to clarify the association between the intensity of intraamniotic inflammatory mediators and cervical *Ureaplasma* spp. colonization in preterm delivery. In addition, it is necessary to investigate the neonatal outcomes associated with *Ureaplasma* spp. infection in preterm delivery.

## 5. Conclusions

Cervical *Ureaplasma* spp. colonization did not affect the intensity of inflammatory mediators in amniotic fluid retrieved during cesarean delivery. Detecting *Ureaplasma* spp. through cervical swab to investigate its effect on intraamniotic inflammation may be negligible. If an intraamniotic infection is suspected, detecting the microorganism through amniocentesis may be necessary.

## Figures and Tables

**Figure 1 ijerph-19-00107-f001:**
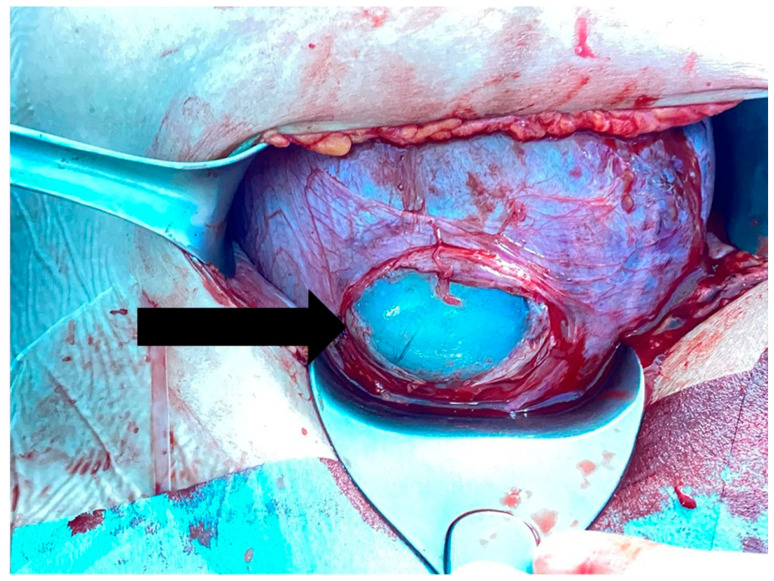
Amniotic membrane (arrow) is exposed after the myometrium is incised with shallow strokes during cesarean section.

**Figure 2 ijerph-19-00107-f002:**
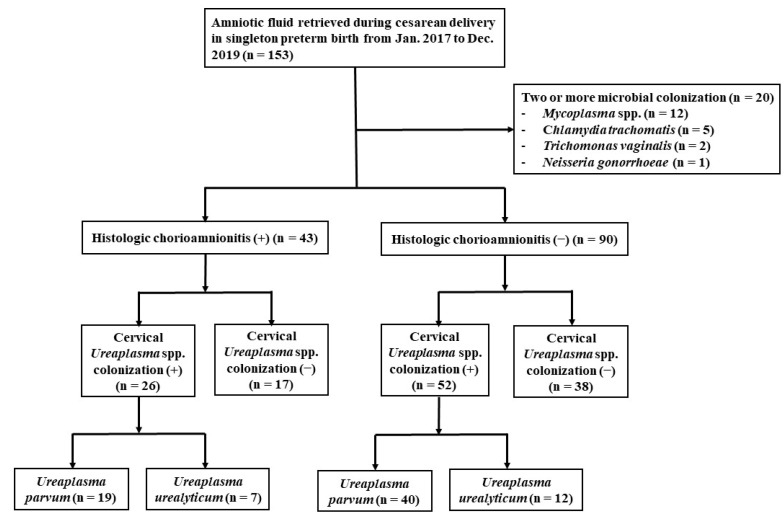
A flowchart of the study.

**Figure 3 ijerph-19-00107-f003:**
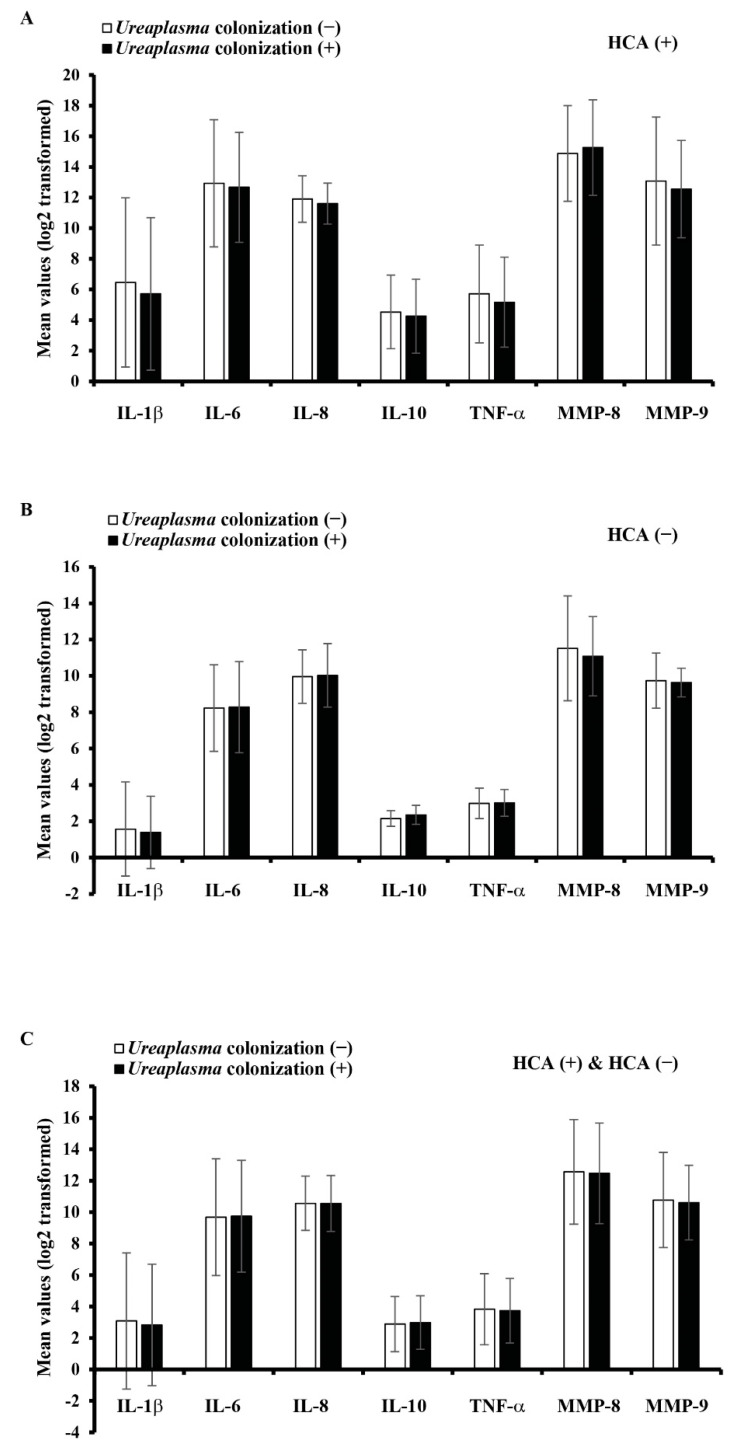
Comparison between the intensity of intraamniotic inflammatory mediators and cervical *Ureaplasma* spp. colonization according to the presence or absence of histologic chorioamnionitis. (**A**) HCA (+), (**B**) HCA (−), and (**C**) HCA (+), HCA (−). All data were not significant (*p* > 0.05). Histologic chorioamnionitis (HCA); Interleukin (IL); Tumor necrosis factor (TNF); Matrix metalloproteins (MMP); Interleukin (IL); Tumor necrosis factor (TNF); Matrix metalloproteins (MMP).

**Figure 4 ijerph-19-00107-f004:**
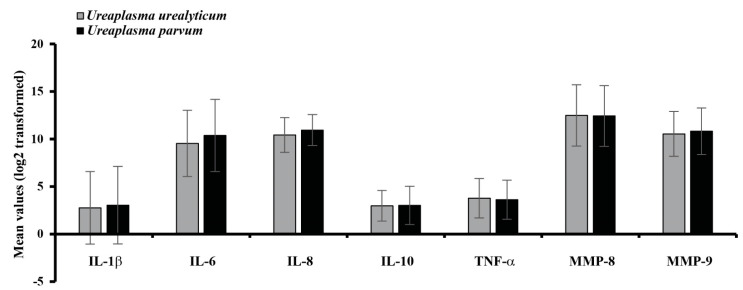
Relative mean values of intraamniotic inflammatory mediators between *Ureaplasma parvum* and *Ureaplasma urealyticum*. Interleukin (IL); Tumor necrosis factor (TNF); Matrix metalloproteins (MMP).

**Table 1 ijerph-19-00107-t001:** A comparison of clinical characteristics and mean values of intraamniotic inflammatory mediators according to the presence or absence of histologic chorioamnionitis.

Variables	Histologic Chorioamnionitis (+) (*n* = 43)	Histologic Chorioamnionitis (−) (*n* = 90)	*p*-Value
Maternal age, years	32.9 ± 4.1	33.4 ± 4.2	0.536
Parity, times	0.6 ± 0.7	0.6 ± 0.7	0.967
Cerclage intervention, *n* (%)	5 (11.6)	9 (10.0)	0.775
Pregnancy-induced hypertension, *n* (%)	11 (25.6)	45 (50.0)	0.008
Gestational diabetes, *n* (%)	4 (9.3)	11 (12.2)	0.619
Cervical dilatation at admission, cm	1.5 ± 1.8	0.8 ± 1.3	0.018
Oligohydramnios, *n* (%)	4 (9.3)	10 (11.1)	0.751
Preterm labor with intact membrane, *n* (%)	22 (51.2)	24 (26.7)	0.005
Premature rupture of membrane, *n* (%)	16 (37.2)	15 (16.7)	0.009
Placenta previa, *n* (%)	3 (7.0)	7 (7.8)	0.870
Placenta abruptio, *n* (%)	2 (4.7)	6 (6.7)	0.647
Cervical *Ureaplasma* spp. colonization, *n* (%)	26 (60.5)	52 (57.8)	0.768
Gestational age at delivery, weeks	30.1 ± 2.5	31.4 ± 2.2	0.008
Birth weight, *g*	1480 ± 472	1616 ± 562	0.170
Apgar score, 1 min	6.1 ± 1.5	6.1 ± 1.6	0.914
Apgar score, 5 min	8.0 ± 0.9	8.0 ± 0.8	0.781
Inflammatory mediators (Logarithmically)			
IL-1 ß	6.0 ± 5.1	1.5 ± 2.3	<0.001
IL-6	12.8 ± 3.8	8.3 ± 2.4	<0.001
IL-8	11.7 ± 1.3	10.0 ± 1.6	<0.001
IL-10	4.4 ± 2.4	2.3 ± 0.5	<0.001
TNF-α	5.4 ± 3.0	3.0 ± 0.8	<0.001
MMP-8	15.1 ± 3.1	10.0 ± 1.6	<0.001
MMP-9	12.8 ± 3.6	9.7 ± 1.1	<0.001

Continuous variables are expressed as mean ± standard deviation. Interleukin (IL); Tumor necrosis factor (TNF); Matrix metalloproteins (MMP).

**Table 2 ijerph-19-00107-t002:** Correlation between intraamniotic inflammatory mediators and gestational age at delivery.

Variables	Gestational Age at Delivery (*n* = 133)	
	Pearson’s Coefficient	*p*-Value
IL-1β	−0.260	0.003
IL-6	−0.176	0.043
IL-8	−0.037	0.672
IL-10	−0.275	0.001
TNF-α	−0.279	0.001
MMP-8	−0.268	0.002
MMP-9	−0.230	0.008

Interleukin (IL); Tumor necrosis factor (TNF); Matrix metalloproteins (MMP).

**Table 3 ijerph-19-00107-t003:** Multivariate logistic regression analysis of the clinical factors in patients with histological chorioamnionitis.

Variables	Odds Ratios	95% Confidence Interval	*p*-Value
Cervical *Ureaplasma* spp. colonization	1.110	0.483–2.550	0.806
Preterm labor with intact membrane	5.041	1.985–12.800	0.001
Premature rupture of membrane	7.217	2.559–20.357	<0.001
Pregnancy-induced hypertension	1.002	0.312–3.217	0.998
Cervical dilatation at admission	1.126	0.854–1.484	0.400
Gestational age at delivery	0.776	0.651–0.926	0.005

## Data Availability

The data presented in this study are available on request from the corresponding author.

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
