# Peer review of "Comparison between Cervical Ureaplasma spp. Colonization and the Intensity of Inflammatory Mediators in the Amniotic Fluid Retrieved during Cesarean Delivery in Preterm Birth"

_ijerph, 2021, doi:10.3390/ijerph19010107_

Round 1

Reviewer 1 Report

Introduction

The choice of inflammatory mediators measured in this study appears to be arbitrary and should be defended.

Results

Please delete horizontal lines for a clearer visualization from Figure 3 and Figure 4.

Please kindly report the R-squared value for Table 2.

Kindly add data about cervical effacement and cervical dilation in Table 1.

Discussion

Relevant findings indicate that NLRP3 inflammasome is a critical complex in the mediation of the inflammatory response, which makes it crucial for the development of pregnancy-induced hypertension and preeclampsia, as well as its complications. Please discuss your results in Table 1 (significant differences in Pregnancy-induced hypertension).

There is evidence suggesting that a sterile anti-inflammatory response is manifested to attenuate the excessive inflammation introduced by low-risk labor at term, involving either the action of a cholinergic pathway, uterine-like myokines or the vaginal microbiome. Some comments are needed for a more integral explanation.

Some authors claim that the systemic downregulation of several members of the IL-10 family of cytokines plays an important role in the activation of myometrial smooth cells associated with uterine contractions during active labor. Some comments are needed for future work.

Author Response

Reviewer #1' comments:

  1. The choice of inflammatory mediators measured in this study appears to be arbitrary and should be defended. -> In keeping with the reviewer’s recommendation, we replaced “inflammatory mediators” with “arbitrarily selected inflammatory mediators” for the purpose of the study in the revised manuscript (Line 24, 60).

  1. Please delete horizontal lines for a clearer visualization from Figure 3 and Figure 4. -> In keeping with the reviewer’s recommendation, we deleted horizontal lines in Figure 3 and 4 of the revised manuscript.
  1. Please kindly report the R-squared value for Table 2. -> In keeping with the reviewer’s recommendation, we have reported the R-squared value in Table 2 of the revised manuscript.
  1. Kindly add data about cervical effacement and cervical dilation in Table 1. -> We are grateful for the reviewer’s thoughtful comments. In keeping with the reviewer’s recommendation, we further investigated cervical ripening at admission. It seemed most appropriate to present the Bishop score including both cervical effacement and cervical dilatation, but it was difficult to obtain the Bishop score retrospectively from the medical record. Therefore, the cervical dilatation was compared and presented in Table 1 and 3 of the revised manuscript. In addition, we have incorporated the relationship between cervical dilatation and HCA in the Methods (Line 51) Results (Line 117-118) and Discussion (Line 181-188) section of the revised manuscript.
  1. Relevant findings indicate that NLRP3 inflammasome is a critical complex in the mediation of the inflammatory response, which makes it crucial for the development of pregnancy-induced hypertension and preeclampsia, as well as its complications. Please discuss your results in Table 1 (significant differences in Pregnancy-induced hypertension). -> We are grateful for the reviewer’s thoughtful comments. we included the paragraph stating that “Meanwhile, in this study, the incidence of pregnancy-induced hypertension in the HCA (+) group was significantly lower than that in the HCA (-) group. Although, the pathophysiology of pregnancy-induced hypertension is still not entirely understood, the development of pregnancy-induced hypertension involves an inflammatory imbalance during placental invasion that leads to endothelial abnormalities [A,B]. According to a large cohort study[C], the lower rate of intrapartum chorioamnionitis in women diagnosed with pregnancy-induced hypertension might be explained by lower proportion of women with pregnancy-induced hypertension presenting to labor and delivery with PPROM.” in the Discussion section of the revised manuscript.

A. Harmon, A. C.; Cornelius, D. C.; Amaral, L. M.; Faulkner, J. L.; Cunningham, M. W., Jr.; Wallace, K.; LaMarca, B., The role of inflammation in the pathology of preeclampsia. Clin Sci (Lond) 2016, 130, 409-419.

B. Socha, M. W.; Malinowski, B.; Puk, O.; Dubiel, M.; Wiciński, M., The NLRP3 Inflammasome Role in the Pathogenesis of Pregnancy Induced Hypertension and Preeclampsia. Cells 2020, 9.

C. Harrison, R. K.; Egede, L. E.; Palatnik, A., Peripartum infectious morbidity in women with preeclampsia. J Matern Fetal Neonatal Med 2021, 34, 1215-1220.

6. There is evidence suggesting that a sterile anti-inflammatory response is manifested to attenuate the excessive inflammation introduced by low-risk labor at term, involving either the action of a cholinergic pathway, uterine-like myokines or the vaginal microbiome. Some comments are needed for a more integral explanation. -> We are grateful for the reviewer’s thoughtful comments. We included the paragraph stating that “Labor is considered as a sterile inflammatory event mainly. The major changes in pro-inflammatory and anti-inflammatory cytokine expression in the cervix occur during the labor process irrespective of gestational age [D,E]. However, previous studies reported that labor was associated with increased risk of microbial invasion of the amniotic cavity, intraamniotic inflammation and HCA [F,G]. This could be explained by the mechanism that unterine contraction had a suction-like effect and that microorganisms sere transferred by digital examination to check the progress of labor [F]. In particular, the intensity of host response to microbial invasion of the amniotic cavity is greater in the context of preterm than term labor [G]. In this study, cervical dilation at admission in the HCA (+) group was also more advanced in singleton preterm births with 24–34 weeks’ gestation.” in the Discussion section of the revised manuscript.

D. Dubicke, A.; Fransson, E.; Centini, G.; Andersson, E.; Byström, B.; Malmström, A.; Petraglia, F.; Sverremark-Ekström, E.; Ekman-Ordeberg, G., Pro-inflammatory and anti-inflammatory cytokines in human preterm and term cervical ripening. J Reprod Immunol 2010, 84, 176-185.

E. Reyes-Lagos, J. J.; Peña-Castillo, M.; Echeverría, J. C.; Pérez-Sánchez, G.; Álvarez-Herrera, S.; Becerril-Villanueva, E.; Pavón, L.; Ayala-Yáñez, R.; González-Camarena, R.; Pacheco-López, G., Women Serum Concentrations of the IL-10 Family of Cytokines and IFN-γ Decrease from the Third Trimester of Pregnancy to Active Labor. Neuroimmunomodulation 2017, 24, 162-170.

F. Seong, H. S.; Lee, S. E.; Kang, J. H.; Romero, R.; Yoon, B. H., The frequency of microbial invasion of the amniotic cavity and histologic chorioamnionitis in women at term with intact membranes in the presence or absence of labor. Am J Obstet Gynecol 2008, 199, 375.e371-375.

G. Romero, R.; Nores, J.; Mazor, M.; Sepulveda, W.; Oyarzun, E.; Parra, M.; Insunza, A.; Montiel, F.; Behnke, E.; Cassell, G. H., Microbial invasion of the amniotic cavity during term labor. Prevalence and clinical significance. J Reprod Med 1993, 38, 543-548.

7. Some authors claim that the systemic downregulation of several members of the IL-10 family of cytokines plays an important role in the activation of myometrial smooth cells associated with uterine contractions during active labor. Some comments are needed for future work. -> We are grateful for the reviewer’s thoughtful comments. We included the paragraph stating that “Labor is considered as a sterile inflammatory event mainly. The major changes in pro-inflammatory and anti-inflammatory cytokine expression in the cervix occur during the labor process irrespective of gestational age [D,E].” in the Discussion section of the revised manuscript.

Reviewer 2 Report

This study investigated whether detection by PCR of Ureaplasma spp in the cervix was associated with the level of inflammatory mediators in the amniotic fluid form Korean women undergoing preterm cesarean delivery. While histologic chorioamnionitis was associated with increased level of all seven inflammatory mediators, detection of cervical Ureaplasma spp was not. These findings support those which find that Ureaplasma is likely not a cause of inflammation leading to chorioamnionitis, but is usually found in the presence of other lower genital tract organisms that are. I have no major concerns with this manuscript. Some minor editing for punctuation and grammar are needed. While likely not significant, did the authors perform stratified analyses i.e. in HCA+ samples did mediator levels differ by Ureaplasma status and similarly in HCA- samples? I think adding this analysis would strengthen the conclusions.

Author Response

This study investigated whether detection by PCR of Ureaplasma spp in the cervix was associated with the level of inflammatory mediators in the amniotic fluid form Korean women undergoing preterm cesarean delivery. While histologic chorioamnionitis was associated with increased level of all seven inflammatory mediators, detection of cervical Ureaplasma spp was not. These findings support those which find that Ureaplasma is likely not a cause of inflammation leading to chorioamnionitis, but is usually found in the presence of other lower genital tract organisms that are. I have no major concerns with this manuscript. Some minor editing for punctuation and grammar are needed. While likely not significant, did the authors perform stratified analyses i.e. in HCA+ samples did mediator levels differ by Ureaplasma status and similarly in HCA- samples? I think adding this analysis would strengthen the conclusions.

Answer: We thank the reviewer for this useful suggestion. In accordance with the reviewer’s recommendation, we have added the stratified analyzes in Figure 3 of the revised manuscript.

Reviewer 3 Report

The authors investigated whether cervical Ureaplasma spp. colonization influences the expression of seven inflammatory mediators in amniotic fluid in singleton preterm birth using RT-PCR.

The research is interesting with certain limitations that should be mentioned.

Did the authors investigate these inflammatory cytokines at protein levels using Immunohistochemistry or ELISA in serum ?

Some of these cytokines, such as IL-8 can have detectable levels in serum.

 If the authors did not use these techniques to confirm their RT-PCR results, they should mention it in the limitations of their study.

Author Response

The authors investigated whether cervical Ureaplasma spp. colonization influences the expression of seven inflammatory mediators in amniotic fluid in singleton preterm birth using RT-PCR.

The research is interesting with certain limitations that should be mentioned.

Did the authors investigate these inflammatory cytokines at protein levels using Immunohistochemistry or ELISA in serum?

Some of these cytokines, such as IL-8 can have detectable levels in serum.

If the authors did not use these techniques to confirm their RT-PCR results, they should mention it in the limitations of their study.

Answer: We are grateful for the reviewer’s thoughtful comments. We analyzed amniotic fluid using a Human Magnetic Luminex screening assay, designed to simultaneously detect and quantitate multiple secreted proteins (e.g., cytokines, chemokines, and growth factors) or expressed genes. This high-throughput technology produces results compared to conventional assays such as ELISA and qPCR with greater efficiency and throughput. Because our data have some limitations to analyzing the causative tissues that secret inflammatory mediators, we described the Discussion section (Line 318-322) of the revised manuscript.

Round 2

Reviewer 1 Report

 The authors have properly addressed all my concerns.